# Using Gas Counter Pressure and Combined Technologies for Microcellular Injection Molding of Thermoplastic Polyurethane to Achieve High Foaming Qualities and Weight Reduction

**DOI:** 10.3390/polym14102017

**Published:** 2022-05-15

**Authors:** Shia-Chung Chen, Kuan-Hua Lee, Che-Wei Chang, Tzu-Jeng Hsu, Ching-Te Feng

**Affiliations:** 1R&D Center for Smart Manufacturing, Chung Yuan Christian University, Taoyuan 32023, Taiwan; mike-0921@hotmail.com (K.-H.L.); landrick0130@gmail.com (C.-W.C.); g10102303@cycu.edu.tw (T.-J.H.); g10873063@cycu.edu.tw (C.-T.F.); 2R&D Center for Semiconductor Carrier, Chung Yuan Christian University, Taoyuan 32023, Taiwan

**Keywords:** microcellular injection molding, thermoplastic polyurethane, injection speed, gas counter pressure, gas holding time, dynamic molding temperature control

## Abstract

Microcellular injection molding technology (MuCell^®^) using supercritical fluid (SCF) as a foaming agent offers many advantages, such as material and energy savings, low cycle time, cost-effectiveness, and the dimensional stability of products. MuCell^®^ has attracted great attention for applications in the automotive, packaging, sporting goods, and electrical parts industries. In view of the environmental issues, the shoe industry, particularly for midsole parts, is also seriously considering using physical foaming to replace the chemical foaming process. MuCell^®^ is thus becoming one potential processing candidate. Thermoplastic polyurethane (TPU) is a common material for molding the outsole of shoes because of its outstanding properties such as hardness, abrasion resistance, and elasticity. Although many shoe manufacturers have tried applying Mucell^®^ processes to TPU midsoles, the main problem remaining to be overcome is the non-uniformity of the foaming cell size in the molded midsole. In this study, the MuCell^®^ process combined with gas counter pressure (GCP) technology and dynamic mold temperature control (DMTC) were carried out for TPU molding. The influence of various molding parameters including SCF dosage, injection speed, mold temperature, gas counter pressure, and gas holding time on the foaming cell size and the associated size distribution under a target weight reduction of 60% were investigated in detail. Compared with the conventional MuCell^®^ process, the implementation of GCP technology or DMTC led to significant improvement in foaming cell size reduction and size uniformity. Further improvement could be achieved by the simultaneous combination of GCP with DMT, and the resulting cell density was about fifty times higher. The successful possibility for the microcellular injection molding of TPU shoe midsoles is greatly enhanced.

## 1. Introduction

Microcellular foaming technology for polymers was proposed and initiated more than three decades ago [1,2]. In 1982, Martini et al. used gas as a blowing agent and developed the foaming of polystyrene (PS) in solid-state in a batch process. In 2001, Trexel Inc. successfully developed a microcellular injection molding process for commercial application and trademarked the process as MuCell^®^ [3] in 2001. The MuCell^®^ process is basically a physical foaming process using supercritical fluids (SCF) as a foaming agent. Compared with conventional injection molding, MuCell^®^ offers many advantages including lower melt viscosity, lower molding pressure, part weight and cycle-time reduction. Foaming also plays a significant role in the packing stage, resulting in shrinkage reduction, warpage minimization, and the elimination of residual stresses, etc. Studies on the process characteristics of MuCell^®^ and the associated molded-part properties have been reported earlier [4,5,6,7,8,9,10,11,12,13,14,15,16,17,18,19,20,21]. Despite its advantages, MuCell^®^ also faces technological difficulties leading to hesitation in its further application. The silver-like swirl flow marks appearing on the molded-part surface as well as the uncontrollable and uneven foaming sizes are the main obstacles that hinder the further application of MuCell^®^. The relevant key issue is believed to be due to the combined effects of the partial foaming and fountain flow effects during the melt-filling process [8].

Recently, the MuCell^®^ process has been extensively considered in the thermoplastic elastomer field [22,23,24,25], mainly for scaffold applications. The foaming skin permits the free transport of nutrients and waste throughout the samples, which is highly desirable in tissue engineering [23]. Another possible application can also be created for shoe soles. The main materials used for shoe soles are thermoplastic polyurethane (TPU), vinyl acetate (EVA), and thermoplastic rubber (TPR) with rubber additives. These materials are wear-resistant, anti-fatigue, light, and flexible and most molding processes are chemical foaming-based methods [26,27]. Due to the environmental issues and regulations, chemical foaming will be more limited. Although TPU [27] has high wear resistance, elasticity, fatigue resistance, chemical resistance, and many other advantages, it has several disadvantages, such as being heavyweight, having high hardness, and having poor damping performance. If the weight reduction offered by the MuCell^®^ process and the overall elasticity of the internal foam can meet the shoe material requirements while maintaining the advantages of TPU, then the MuCell process for TPU may become one of the new solutions for the shoe industry. From the shoe industry’s view point, the weight reduction ratio, part elasticity, and hardness are the major concerns. All these are intimately related to the foaming cell qualities. A high foaming density, fine cell size, and uniform size distribution seem to be required to in order to fulfill the formed sole properties. Thus, in this study we extend our foaming control experiences in MuCell^®^ of thermoplastics [17,18,19,20,21], namely, gas counter pressure and mold-temperature control constituting the so-called P (pressure)-T (temperature) path [21], to see if it also works for thermoplastic elastomers (such as TPU). Relevant processing parameters including SCF dosage, injection speed, mold temperature, gas counter pressure, and gas holding time on foaming cell size and the associated size distribution were investigated in detail. Dynamic mold-temperature was also implemented and employed with GCP (gas counter pressure); the corresponding foaming qualities were examined as well.

## 2. Experimental Procedure

### 2.1. Foaming Materials

The material used in this experiment was Elastollan^®^ 1185A thermoplastic polyurethane resin (TPU) from BASF (Ansan, Korea). Its property advantages of high strength, toughness, and wear resistance have attracted the shoe manufacturers. Its recommended processing temperature is 190–220 °C and the relevant physical properties are given in Table 1.

### 2.2. MuCell^®^ Injection Molding Machine and Gas Counter Pressure Regulation

The injection machine used was an ARBURG 420C MuCell^®^ Injection machine (Wilmington, MA, USA). It has a screw diameter of 40 mm and a maximum clamping force of 1000 kN. The supercritical fluid (SCF) generator, made by Trexel, uses nitrogen as a foaming agent. SCF was injected into the barrel and well mixed and dissolved within the melt. A homemade gas-pressure-regulation unit installed with a high-frequency gas-control valve was utilized. Dynamic mold temperature was carried out using high- and low-temperature (DMTC) units by switchover in a designed sequence. A schematic can be referred to in Figure 3 of Ref. [21].

### 2.3. Experimental Mold

The mold used in this study was a plate-part model, with a fan gate and thickness identical to that of the product. The product size was 170 × 40 × 6 mm. It had five straight symmetrical cooling channels on both the core and cavity sides (Figure 1). Three locations for foaming morphology examinations are also indicated. The gas inlet/outlet was mounted at the far-gate site of the product.

DMTC was applied through two mold-temperature controllers, keeping at the target high and low temperatures, and connected together with valves that could switch the manifold from one to another. Once the melt was filled (including holding time in GCP cases), the mold-temperature unit was switched over from the high coolant temperature to the lower coolant temperature. Prior to the beginning of the melt filling for the next cycle, the mold-temperature unit was set to high temperature again. All of the cooling channels can raise or lower the coolant temperature together.

### 2.4. Electronic Balancer

An LWL precision analytical balance (LB-201S) was used to measure the molded-part weight. The maximum weight was 210 g with an accuracy of ±0.0001 g.

### 2.5. Scanning Electron Microscope

This study used Hitachi’s scanning electron microscope (S-3000N) with a magnification between 5 times and 300000 times. The test specimen was cut at three locations, P1, P2, and P3. The section surfaces were covered with a layer of gold foil via sputtering and the associated foaming morphologies were examined. The average foaming size, as well as cell density, could then be calculated. A typical scanning electron microscope (SEM) image for measuring foaming bubbles (halfway across the thickness section) is presented in Figure 2.

### 2.6. Experimental Parameters

The research was divided into four parts, so as to facilitate readers to find the experimental results. The first part investigated the influences of molding parameters (mold temperature, injection speed, and supercritical fluid dosage) on the foaming cell characteristics of the TPU product. In this part of the experiment, the weight reduction of the product was fixed at 60%. The three parameter values including their combinations are given in Table 2. In the second part, the gas counter pressure and gas holding time were applied to investigate the impact of the cell uniformity and size. The relevant parameters for gas counter pressure and molding conditions are listed in Table 3. Thirdly, the DMTC was carried out alone with the traditional MuCell^®^ process to study its influence. Finally, GCP simultaneously combined with DMTC was employed to optimize the foaming qualities. If a too high GCP value was applied, the molded parts were short shot and 60% weight reduction could not be achieved. The process conditions of Table 2 and Table 3 guarantee the target weight reduction of 60%.

### 2.7. Cell Size and Density Measurement

In this study, the average size of foaming cells, foaming cell density, and the cell diameter distribution were calculated from the foaming morphologies. The molded products were cut into three sections, namely, the front, center, and rear portions. The cross-sections of the SEM photographs for morphologies were examined and measured location (P1, P2, P3), as shown in Figure 2. Based on the distribution of local cells, the SEM image was analyzed in an AutoCAD and cell size and density calculated. The number of cells per unit volume can be calculated based on the cell density formula listed below:$$N_{0}={\left(\frac{nm^{2}}{A}\right)}^{\frac{3}{2}}\times \frac{1}{1-\sum \frac{4}{3}\pi R_{I}{}^{3}}$$
$$={\left(\frac{nm^{2}}{A}\right)}^{\frac{3}{2}}\times \frac{1}{1-\left(\frac{4}{3}\pi R_{avg}{}^{3}\times {\left(\frac{nm^{2}}{A}\right)}^{\frac{3}{2}}\right)}$$
where *n* is the number of cells, *A* is the measurement area, *m* is the magnification, *R_avg_* is the average diameter of the cell, and *N*_0_ is the density of the foam.

## 3. Results and Discussion

In a traditional MuCell^®^ process for TPU molding, the larger-size cells appear at the rear end of the flow direction due to the lower pressure around the melt front. The relatively higher pressure near the gate results in smaller and more uniform cell sizes.

The combination of different processing parameters creates quite a lot of data; therefore, only selected illustrations are shown to describe the relevant influences.

### 3.1. Traditional MuCell Process

The effect of mold temperature on foaming morphology is shown in Figure 3. The higher the mold temperature, the larger the cell size and the lower the average cell density, as seen in Table 4. In the typical MuCell^®^ process, the maximum size of the cell usually appears in the core layer and it becomes smaller as it approaches layers near the mold wall. Both the cell sizes near the frozen layer and near the core layer increase with increased mold temperature. Figure 4a shows the cell size variations along the gap-wise direction (0 at skin and 3 at core) at location P3 and the associated cell-size distributions are depicted in Figure 4b. It can be seen that the average size of the cell near the skin region gradually increases from 26.34 μm at 30 °C, to 213.27 μm at 90 °C. It is clear that a higher temperature favors the large-size cell formation. The results for the averaged cell densities are shown in Table 4. Higher cavity pressure and lower mold temperature favor cell nucleation and thus led to higher cell density at a mold temperature of 30 °C at the near-gate location, P1. The higher cell density also results in a higher percentage of smaller cell sizes. To reduce the length of the article, the detailed cell-size distribution information will not be shown for the influence from other processing parameters until the last optimized case.

Regarding the influence of injection speed, the foaming morphology and the averaged cell densities are given in Figure 5 and Table 5, respectively. The higher injection speeds lead to higher cavity pressure and, thus, favor higher cell density formation. However, its influence is less significant than mold temperature.

For the influence of different SCF dosages, the results in Figure 6 and Table 6 indicate that higher SCF dosages favor high cell density formation. When the SCF dosage is increased from 0.4 wt% to 1.0 wt%, the density of the foam increases from 2.70 × $10^{11}$ cell/cm^3^ to 9.26 × $10^{11}$ cell/cm^3^. This is because a higher SCF dosage can help the cell nucleation; thus, the nucleation density increases.

### 3.2. GCP-Employed MuCell Process

The influence of GCP on foaming morphology can be seen in Figure 7 and the associated cell densities under different GCP values and at various locations are illustrated in Table 7. The GCP does restrict the foaming in the filling stage and leads to more significant nucleation in the post-filling process, and thus, ends up with higher cell density. The employment of GCP results in a cell density increase of about 3~5 times. The influence of gas pressure holding time on cell sizes and densities can be found in Figure 8 and Table 8, respectively. A high holding time tends to favor high cell density; however, when the holding time is longer than 25 s, the influence becomes not obvious.

### 3.3. DMTC Involed MuCell Proccess

The dynamic mold temperature in this study basically determined the cooling rate (Table 3), a fixed mold temperature was used in the experiment (Table 5), the corresponding influence is shown in Figure 9, and the relevant cell densities are listed in Table 9. Basically, a slower cooling rate tends to favor high cell density formation. Before optimizing the process parameters for foaming quality control, let us first compare the progressive results among traditional MuCell^®^, MuCell^®^ combined with GCP and MuCell^®^ plus DMTC. Figure 10a shows the corresponding morphologies at P1, P2, and P3 locations. The cell size variations across the thickness section (P1 location) are shown in Figure 10b. In general, MuCell^®^ combined with either GCP or DMT results in better foaming qualities in both cell-density and cell-size distributions. However, the individual influences from DMTC and GCP seem to be situation-dependent. Table 10 shows effectiveness compare of improving foaming qualities between MuCell combined with either GCP or DMTC. Both group perform better in both cell density and cell size distributions, but DMTC a little better.

### 3.4. GCP and DMTC Combined MuCell Process

Finally, the foaming qualities of MuCell combined with both GCP and DMTC were compared with the traditional MuCell^®^ process. The results are illustrated in Figure 11a–g, and the associated cell densities at various locations are listed in Table 11. The averaged cell density distribution at P1, P2, and P3 along the flow direction is also shown in Figure 11. Obviously, the GCP-based combination technology leads to significant improvement in foaming qualities including fine cell sizes, more uniform cell-size distribution and much higher cell density.

## 4. Conclusions

In this study, the MuCell^®^ process combined with gas counter pressure (GCP) technology and dynamic mold temperature control were carried out for TPU molding. The influence of various molding parameters, including SCF dosage, injection speed, mold temperature, gas counter pressure, and gas holding time on foaming cell size and the associated size distribution under a target of 60% weight reduction were investigated in detail.

An increase in mold temperature seems to favor the foaming cell growth, and as a result, the averaged cell size also increases on both skin and core layers. To the contrary, higher mold temperature reduces the overall cell density. Increased injection speed was found to reduce the cell diameter. Higher injection speed accompanies higher injection pressure, leading to smaller cell sizes and higher cell densities. Increasing the SCF dosage increases the foaming nuclei number and reduces the overall cell sizes. Moreover, an increase in the SCF dosage also helps to increase the inner pressure between the cells, which will restrict large cells foaming at the rear end of flow direction. Gas counter pressure can effectively inhibit forming during the melt-filling stage, leading to smaller cell sizes and higher cell densities. Moreover, GCP has a more significant influence on cell foaming at the P3 location. High holding time tends to favor high cell density, particularly in the rear end site; however, when the holding time is longer than 25 s, its influence becomes non-obvious. As far as the influence of dynamic mold-temperature variation, the slower cooling rate favors the formation of smaller cell size and high-density formation. Both the individual effects of GCP and DMTC alone with MuCell^®^ show a significant influence on cell density and cell size reduction in an approximately equal manner. Finally, the simultaneous combination of GCP and DMTC led to the highest cell densities, smallest cell sizes and best uniformity in cell-size distribution. As seen from Table 11, the cell densities are more than fifty times higher when compared with conventional MuCell^®^. This is again consistent with the P (pressure)-T (temperature) path control concept for MuCell^®^ foaming in thermoplastics proposed and published earlier. The application possibility for the microcellular injection molding of TPU shoe midsoles is greatly enhanced.

## Figures and Tables

**Figure 1 polymers-14-02017-f001:**
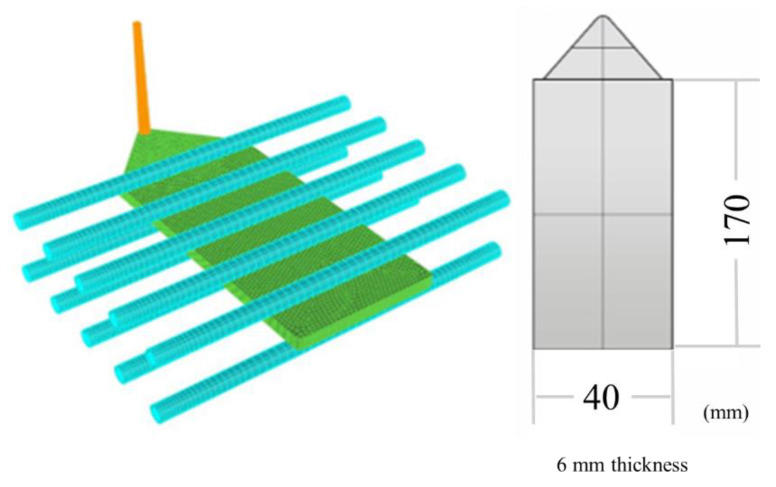
Plate-part mold used for molding experiments.

**Figure 2 polymers-14-02017-f002:**
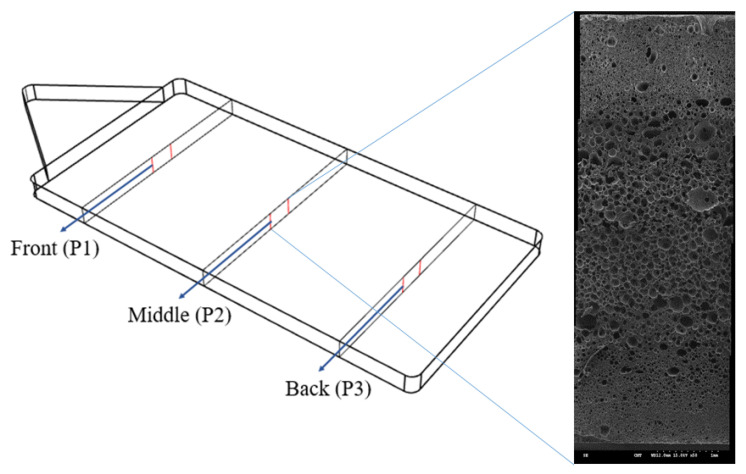
A typical SEM image for measuring foaming bubbles photographed at P1, P2, and P3. A typical foaming morphology at P2 location is illustrated.

**Figure 3 polymers-14-02017-f003:**
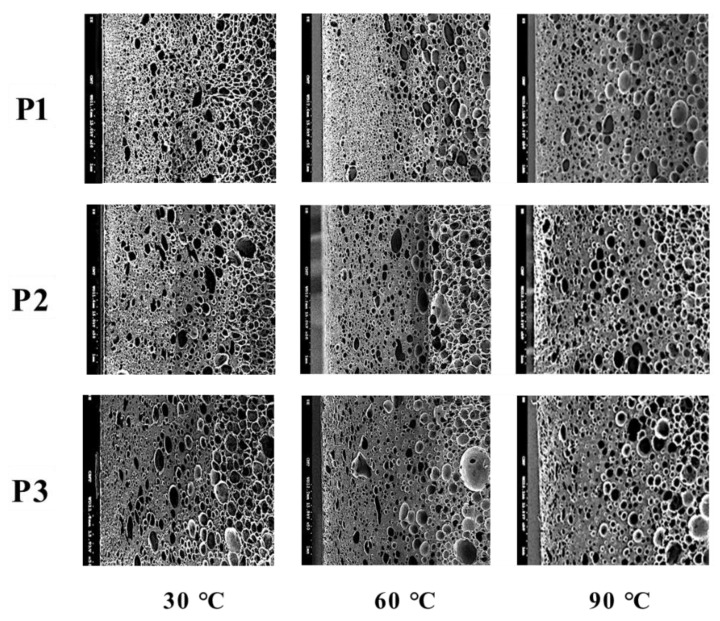
The effect of mold temperature on foaming morphology at various locations.

**Figure 4 polymers-14-02017-f004:**
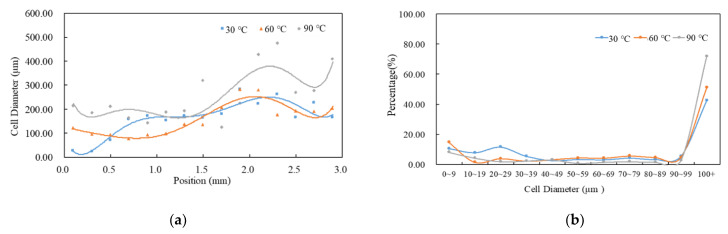
Distribution of cell sizes under different mold temperatures at P3. (**a**) Cell diameter distribution across the thickness section. Here, 0 represents part skin location and 3 represents thickness center location. (**b**) Overall cell diameter distribution.

**Figure 5 polymers-14-02017-f005:**
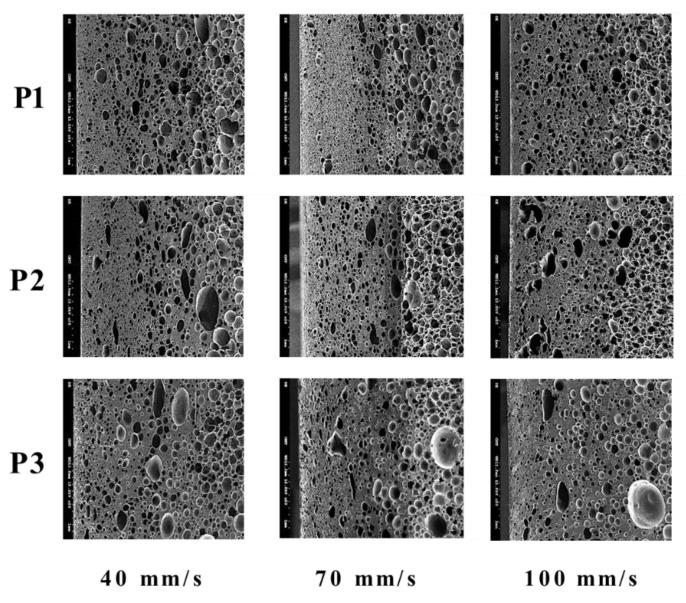
The effect of injection speed on foaming morphology.

**Figure 6 polymers-14-02017-f006:**
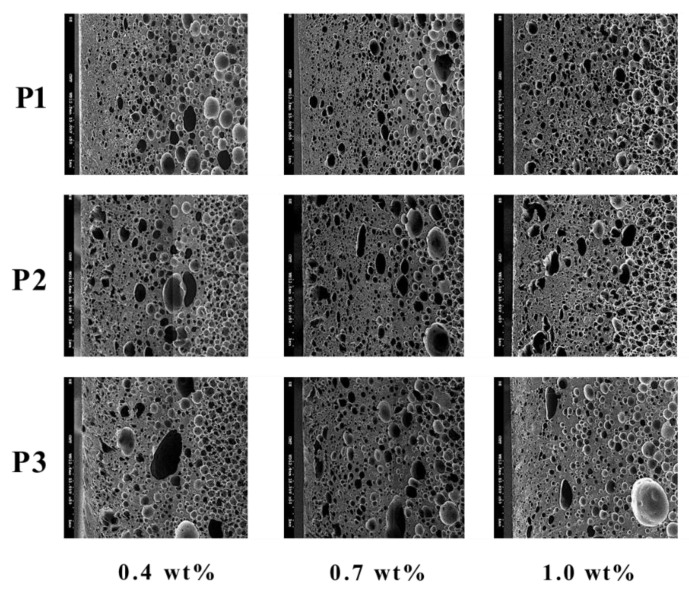
The effect of different SCF dosages on foaming morphology.

**Figure 7 polymers-14-02017-f007:**
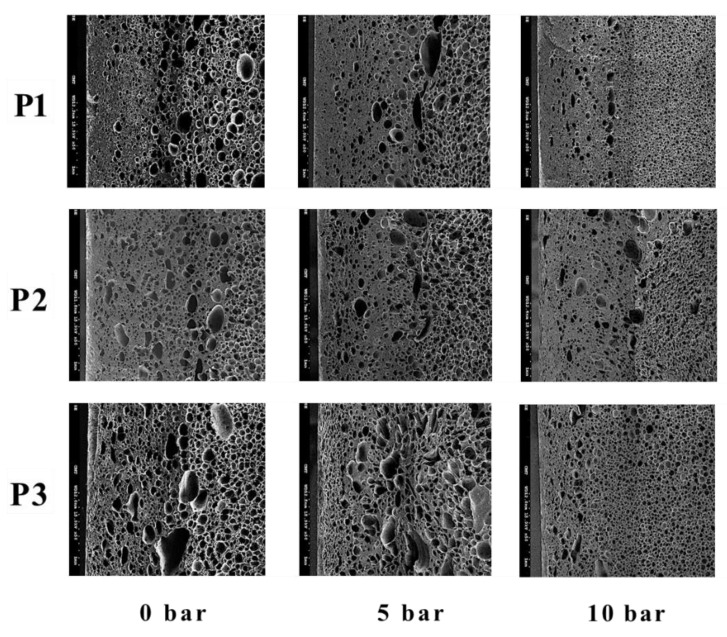
The effect of different GCP pressures on foaming morphology.

**Figure 8 polymers-14-02017-f008:**
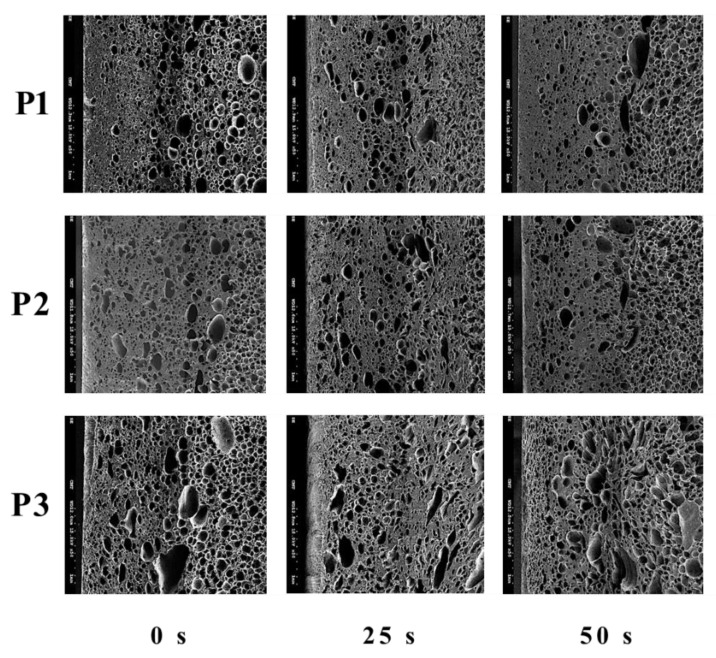
The effect of different GCP holding times on foaming morphology.

**Figure 9 polymers-14-02017-f009:**
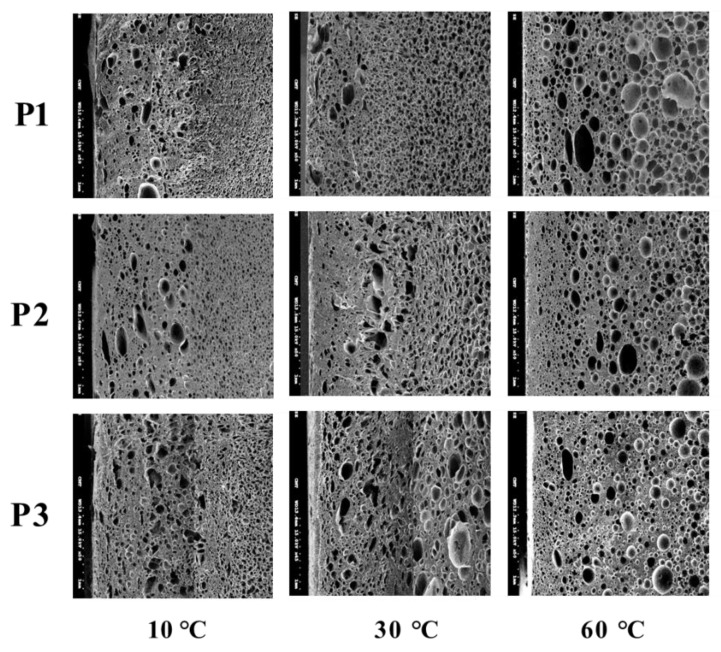
The effect of DMTC corresponding to different cooling rates on foaming morphology.

**Figure 10 polymers-14-02017-f010:**
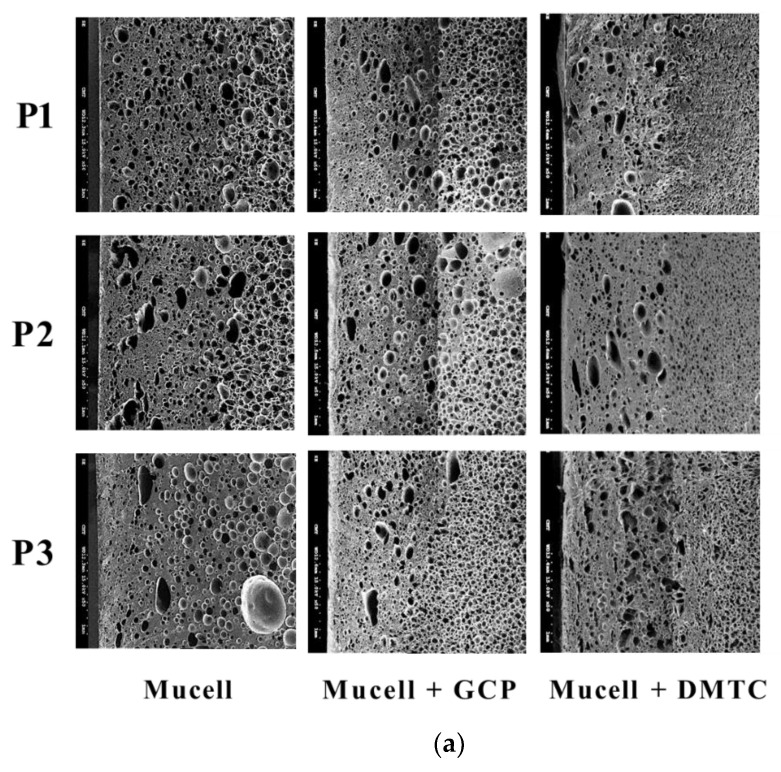
Comparison of foaming morphology among traditional MuCell^®^, MuCell plus GCP and MuCell plus DMTC. (**a**) SEM photograph. (**b**) Cell diameter across the section at P1. (**c**) Cell-diameter distribution at P1.

**Figure 11 polymers-14-02017-f011:**
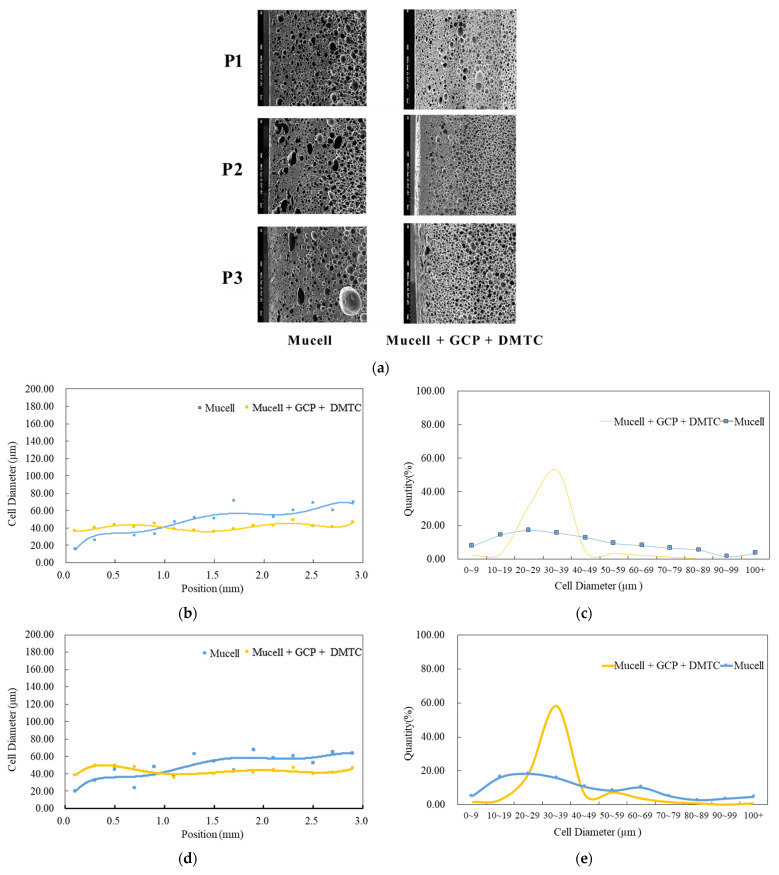
Comparison of foaming morphologies between traditional MuCell^®^ and its simultaneous combination with GCP and DMTC. (**a**) SEM photograph. (**b**) Cell diameter variations across section at P1. (**c**) Cell diameter distribution at P1. (**d**) Cell diameter variations across section at P2. (**e**) Cell diameter distribution at P2. (**f**) Cell diameter variations across section at P3. (**g**) Cell diameter distribution at P3.

**Table 1 polymers-14-02017-t001:** Properties of BASF 1185A TPU.

Properties	1185A	Units
Density	1.12	$\mathrm{cells}/{\mathrm{cm}}^{3}$
Hardness	85	Shore A
Tensile Strength	45	MPa
Mold Temp.	15~70	°C
Melt Temp.	190~220	°C
Vicar Softening Point	85	°C

**Table 2 polymers-14-02017-t002:** Processing parameters (traditional MuCell^®^).

No.	Weight Reduction (%)	SCF Dosage (wt%)	Mold Temp. (°C)	Injection Speed (mm/s)
1	60	0.4	60	70
2		0.7	30	
3			90	
4			60	40
5				70
6				100
7		1.0		70

**Table 3 polymers-14-02017-t003:** Processing parameters (with GCP and/or DMTC).

SCF Dosage (wt%)	0.4, 0.7, 1.0
Mold Temperature (°C)	30, 60, 90
Injection Speed (mm/sec)	40, 70, 100
Gas Counter Pressure (bar)	5 and 10
GCP Holding Times (second)	5, 25, 50
Dynamic Mold Temperature Variation Range (Cooling Rate)	120 °C to 60 °C (3.4 °C/s) 120 °C to 30 °C (5.5 °C/s) 120 °C to 10 °C (7.8 °C/s)

**Table 4 polymers-14-02017-t004:** Averaged cell density at different mold temperatures.

Mold Temp.	P1	P2	P3
30 °C	6.91 × $10^{10}$ cells/${\mathrm{cm}}^{3}$	5.43 × $10^{10}$ cells/${\mathrm{cm}}^{3}$	5.43 × $10^{10}$ cells/${\mathrm{cm}}^{3}$
60 °C	4.70 × $10^{10}$ cells/${\mathrm{cm}}^{3}$	4.76 × $10^{10}$ cells/${\mathrm{cm}}^{3}$	4.04 × $10^{10}$ cells/${\mathrm{cm}}^{3}$
90 °C	3.06 × $10^{10}$ cells/${\mathrm{cm}}^{3}$	1.73 × $10^{10}$ cells/${\mathrm{cm}}^{3}$	1.73 × $10^{10}$ cells/${\mathrm{cm}}^{3}$

**Table 5 polymers-14-02017-t005:** Averaged cell density at different injection speeds.

Injection Speed	P1	P2	P3
40 mm/s	$1.89\;\times \;10^{11}$ $\;\mathrm{cells}/{\mathrm{cm}}^{3}$	$1.72\;\times \;10^{11}$ $\mathrm{cells}/{\mathrm{cm}}^{3}$	$5.64\;\times \;10^{11}$ $\;\mathrm{cells}/{\mathrm{cm}}^{3}$
70 mm/s	$1.34\;\times \;10^{12}$ $\;\mathrm{cells}/{\mathrm{cm}}^{3}$	$1.29\;\times \;10^{12}$ $\;\mathrm{cells}/{\mathrm{cm}}^{3}$	$1.30\;\times \;10^{12}$ $\;\mathrm{cells}/{\mathrm{cm}}^{3}$
100 mm/s	$3.06\;\times \;10^{12}$ $\;\mathrm{cells}/{\mathrm{cm}}^{3}$	$1.73\;\times \;10^{12}$ $\;\mathrm{cells}/{\mathrm{cm}}^{3}$	$1.73\;\times \;10^{12}$ $\;\mathrm{cells}/{\mathrm{cm}}^{3}$

**Table 6 polymers-14-02017-t006:** Averaged cell density at different SCF dosages.

SCF Dosage	P1	P2	P3
0.4 wt%	$2.70\;\times \;10^{11}$ $\;\mathrm{cells}/{\mathrm{cm}}^{3}$	$2.8\;\times \;10^{11}$ $\;\mathrm{cells}/{\mathrm{cm}}^{3}$	$1.66\;\times \;10^{11}$ $\;\mathrm{cells}/{\mathrm{cm}}^{3}$
0.7 wt%	$2.80\;\times \;10^{11}$ $\;\mathrm{cells}/{\mathrm{cm}}^{3}$	$6.28\;\times \;10^{11}$ $\;\mathrm{cells}/{\mathrm{cm}}^{3}$	$3.02\;\times \;10^{11}$ $\;\mathrm{cells}/{\mathrm{cm}}^{3}$
1.0 wt%	$9.26\;\times \;10^{11}$ $\;\mathrm{cells}/{\mathrm{cm}}^{3}$	$7.93\;\times \;10^{11}$ $\;\mathrm{cells}/{\mathrm{cm}}^{3}$	$8.09\;\times \;10^{11}$ $\;\mathrm{cells}/{\mathrm{cm}}^{3}$

**Table 7 polymers-14-02017-t007:** Averaged cell density at different GCP pressures (SCF 0.7%wt, GCP holding time 50 s).

GCP Pressure	P1	P2	P3
0 bar	$7.21\;\times \;10^{11}$ $\;\mathrm{cells}/{\mathrm{cm}}^{3}$	$6.28\;\times \;10^{11}$ $\;\mathrm{cells}/{\mathrm{cm}}^{3}$	$6.02\;\times \;10^{11}$ $\;\mathrm{cells}/{\mathrm{cm}}^{3}$
5 bar	$1.57\;\times \;10^{12}$ $\;\mathrm{cells}/{\mathrm{cm}}^{3}$	$1.27\;\times \;10^{12}$ $\;\mathrm{cells}/{\mathrm{cm}}^{3}$	$9.34\;\times \;10^{11}$ $\;\mathrm{cells}/{\mathrm{cm}}^{3}$
10 bar	$3.69\;\times \;10^{12}$ $\;\mathrm{cells}/{\mathrm{cm}}^{3}$	$2.89\;\times \;10^{12}$ $\;\mathrm{cells}/{\mathrm{cm}}^{3}$	$2.05\;\times \;10^{12}$ $\;\mathrm{cells}/{\mathrm{cm}}^{3}$

**Table 8 polymers-14-02017-t008:** Averaged cell density at different GCP holding times (SCF 0.7%wt, GCP 5 bar).

Holding Time	P1	P2	P3
5 s	$1.60\;\times \;10^{12}$ $\;\mathrm{cells}/{\mathrm{cm}}^{3}$	$1.43\;\times \;10^{12}$ $\;\mathrm{cells}/{\mathrm{cm}}^{3}$	$7.95\;\times \;10^{11}$ $\;\mathrm{cells}/{\mathrm{cm}}^{3}$
25 s	$1.57\;\times \;10^{12}$ $\;\mathrm{cells}/{\mathrm{cm}}^{3}$	$1.36\;\times \;10^{12}$ $\;\mathrm{cells}/{\mathrm{cm}}^{3}$	$8.57\;\times \;10^{11}$ $\;\mathrm{cells}/{\mathrm{cm}}^{3}$
50 s	$1.57\;\times \;10^{12}$ $\;\mathrm{cells}/{\mathrm{cm}}^{3}$	$1.27\;\times \;10^{12}$ $\;\mathrm{cells}/{\mathrm{cm}}^{3}$	$9.34\;\times \;10^{11}$ $\;\mathrm{cells}/{\mathrm{cm}}^{3}$

**Table 9 polymers-14-02017-t009:** Averaged cell density at different DMTC situations (cooling rate).

Varied Mold Temperature	P1	P2	P3
120–60 °C (3.4 °C/s)	$1.83\;\times \;10^{12}$ $\;\mathrm{cells}/{\mathrm{cm}}^{3}$	$1.89\;\times \;10^{12}$ $\;\mathrm{cells}/{\mathrm{cm}}^{3}$	$1.78\;\times \;10^{12}$ $\;\mathrm{cells}/{\mathrm{cm}}^{3}$
120–30 °C (5.5 °C/s)	$1.10\;\times \;10^{12}$ $\;\mathrm{cells}/{\mathrm{cm}}^{3}$	$8.25\;\times \;10^{11}$ $\;\mathrm{cells}/{\mathrm{cm}}^{3}$	$7.62\;\times \;10^{11}$ $\;\mathrm{cells}/{\mathrm{cm}}^{3}$
120–10 °C (7.8 °C/s)	$1.08\;\times \;10^{12}$ $\;\mathrm{cells}/{\mathrm{cm}}^{3}$	$7.20\;\times \;10^{11}$ $\;\mathrm{cells}/{\mathrm{cm}}^{3}$	$6.62\;\times \;10^{11}$ $\;\mathrm{cells}/{\mathrm{cm}}^{3}$

**Table 10 polymers-14-02017-t010:** Averaged cell densities among traditional MuCell^®^ and its combination with GCP and DMTC alone.

Process Type	P1	P2	P3
Mucell^®^	$4.70\;\times \;10^{10}$ $\;\mathrm{cells}/{\mathrm{cm}}^{3}$	$4.76\;\times \;10^{10}$ $\;\mathrm{cells}/{\mathrm{cm}}^{3}$	$4.04\;\times \;10^{10}$ $\;\mathrm{cells}/{\mathrm{cm}}^{3}$
Mucell^®^ + GCP (SCF1.0%wt, GCP10 bar, 50 s holding time)	$2.08\;\times \;10^{12}$ $\;\mathrm{cells}/{\mathrm{cm}}^{3}$	$2.00\;\times \;10^{12}$ $\;\mathrm{cells}/{\mathrm{cm}}^{3}$	$2.17\;\times \;10^{12}$ $\;\mathrm{cells}/{\mathrm{cm}}^{3}$
Mucell^®^ + DMTC 120–60 °C (7.8 °C/s)	$1.83\;\times \;10^{12}$ $\;\mathrm{cells}/{\mathrm{cm}}^{3}$	$1.89\;\times \;10^{12}$ $\;\mathrm{cells}/{\mathrm{cm}}^{3}$	$1.78\;\times \;10^{12}$ $\;\mathrm{cells}/{\mathrm{cm}}^{3}$

**Table 11 polymers-14-02017-t011:** Averaged cell density comparison between traditional MuCell^®^ and its simultaneous combination with GCP and DMTC.

Process Type	P1	P2	P3
Mucell^®^	$4.70\;\times \;10^{10}$ $\;\mathrm{cells}/{\mathrm{cm}}^{3}$	$4.76\;\times \;10^{10}$ $\;\mathrm{cells}/{\mathrm{cm}}^{3}$	$4.04\;\times \;10^{10}$ $\;\mathrm{cells}/{\mathrm{cm}}^{3}$
Mucell^®^ + GCP + DMTC	$2.85\;\times \;10^{12}$ $\;\mathrm{cells}/{\mathrm{cm}}^{3}$	$2.77\;\times \;10^{12}$ $\;\mathrm{cells}/{\mathrm{cm}}^{3}$	$2.69\;\times \;10^{12}$ $\;\mathrm{cells}/{\mathrm{cm}}^{3}$

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
