# Peer review of "Using Gas Counter Pressure and Combined Technologies for Microcellular Injection Molding of Thermoplastic Polyurethane to Achieve High Foaming Qualities and Weight Reduction"

_polymers, 2022, doi:10.3390/polym14102017_

Round 1

Reviewer 1 Report

This manuscript explores the effects of gas counter pressure and dynamic mold temperature control on Microcellular injection molding. Compared to the previous article, TPU materials are used in this manuscript, and it may provide a reference for related fields. We recommend that this paper can be published after revision as indicated.

  • Line 98, Dynamic Mold Temperature control should be introduced here.
  • Line 121, “The research is divided into four parts”. Is it possible to introduce the RESULTS in sub-headings, so as to facilitate readers to find the experimental results?
  • Line 185, In Table 5, a fixed mold temperature is used in the experiment. What is “Dynamic mold temperature”?
  • Line 206, figure 4(a). Please state that "position" refers to sample thickness.
  • Line 243, Figure11, Cell density is very important, but why is there no “Cell density Vs. position” in figures?

Author Response

Line 98, Dynamic Mold Temperature control should be introduced here.

【R】The statement is added: Once the melt was filled (including holding time in GCP cases), the mold temperature unit was switchover from the high coolant temperature to the lower coolant temperature. Prior to the beginning of the melt filling for the next cycle, the mold temperature unit was set to high temperature again.

Line 121, “The research is divided into four parts”. Is it possible to introduce the RESULTS in sub-headings, so as to facilitate readers to find the experimental results?

【R】Thanks for the suggestion! Sub-heading was included.

Line 185, In Table 5, a fixed mold temperature is used in the experiment. What is “Dynamic mold temperature”?

【R】In column 6 of Table 3, three DMTC cases were described. The results of DMTC cases were shown in Table 9. Cases of Table 4 to Table 8 were the results of fixed mold temperature but may have different values.

Line 206, figure 4(a). Please state that "position" refers to sample thickness.

【R】The statement is added. 0 represents part surface location, 3 is the part center along thickness direction.  

Line 243, Figure11, Cell density is very important, but why is there no “Cell density Vs. position” in figures?

【R】Position dependence of cell sizes in the thickness direction were shown in Figs. 4 to 11. Cell density has a statistical feature, therefore, the averaged cell density is shown only in the melt flow direction (P1 to P3 in Tables 4 to 11).

Reviewer 2 Report

Review Comments

This paper reported a study on the microcellular injection molding of thermoplastic polyurethane (TPU) using gas counter pressure (GCP), dynamic mold temperature control (DMTC), and microcellular injection molding (MuCell) techniques. The effect of various processing parameters on the cell morphologies were examined. The experimental results presented here are sufficient and convincing. Most importantly, it was found that the cell densities of TPU samples produced by combining GCP, DMTC, and MuCell techniques are more than 50 times higher than those produced by the conventional MuCell process. Therefore, this paper is suitable for being accepted and published in Polymers. The followings are some suggestions and comments that could further improve the readability and completeness of this paper.

  1. Page 2, Line 48 - It’s not recommended to cite 18 references in one sentence. You will likely need to rewrite your citation sentences to discuss how these citations contribute to your research field.
  2. Figure 1 - The dimensions in Figure 1 are not consistent with that described in Page 3, Line 11.
  3. Page 5, Line 144 – It is stated that “P is the measurement area”. It should be A, instead of P.
  4. Page 5, Line 160 – It is stated that “It can be seen that the average size of the cell near the core center gradually increases from 26.34 μm at 30°C to 213.27 μm at 90°C.” Please double check whether it is the core center or surface layer.
  5. Figure 3, 5-11 - The magnification and scale bar for the SEM images are too small to read. Please add legible scale bars to the SEM images.
  6. Tables 4-11 – Please add the unit of cell density to these tables.
  7. There are some grammar errors and spelling mistakes as listed below.
  8. Page 1, Line 25 – “were carried out to for TPU molding” should be “were carried out for TPU molding”  
  9. Page 1, Line 37 – “Microcellular foaming technology or polymers was proposed” should be “Microcellular foaming technology for polymers was proposed”
  10. Page 2, Line 70 – “A high forming density” should be “A high foaming density”
  11. Page 4, Line 129 – “Finally, Finally,” should be “Finally”
  12. Page 5, Line 155 – Please rewrite “the maximum size of the cell usually appears in the core layer becomes smaller as the layer approaches the mold wall” As “the maximum size of the cell usually appears in the core layer and it becomes smaller as it approaches layers near the mold wall”.

Author Response

Page 2, Line 48 - It’s not recommended to cite 18 references in one sentence. You will likely need to rewrite your citation sentences to discuss how these citations contribute to your research field.

【R】Thanks for the suggestion. The references has been separately describe in later paragraph.

Figure 1 - The dimensions in Figure 1 are not consistent with that described in Page 3, Line 11.

【R】Thanks for pointing out error. Fig. 1 has been corrected.

Page 5, Line 144 – It is stated that “P is the measurement area”. It should be A, instead of P.

【R】Thanks for pointing out error. The error has been corrected.

Page 5, Line 160 – It is stated that “It can be seen that the average size of the cell near the core center gradually increases from 26.34 μm at 30°C to 213.27 μm at 90°C.” Please double check whether it is the core center or surface layer.

【R】Thanks for pointing out error. It should be the reverse from skin to core. The error has been corrected.

Figure 3, 5-11 - The magnification and scale bar for the SEM images are too small to read. Please add legible scale bars to the SEM images.

【R】More clear pictures were provided. The legible scale bars were also indicated in previous letter.

Tables 4-11 – Please add the unit of cell density to these tables.

【R】Thank! We already added the unit in all tables as suggested.

There are some grammar errors and spelling mistakes as listed below.

Page 1, Line 25 – “were carried out to for TPU molding” should be “were carried out for TPU molding” 

Page 1, Line 37 – “Microcellular foaming technology or polymers was proposed” should be “Microcellular foaming technology for polymers was proposed”

Page 2, Line 70 – “A high forming density” should be “A high foaming density”

Page 4, Line 129 – “Finally, Finally,” should be “Finally”

【R】Thanks for pointing out error. The typing error has been corrected.

Page 5, Line 155 – Please rewrite “the maximum size of the cell usually appears in the core layer becomes smaller as the layer approaches the mold wall” As “the maximum size of the cell usually appears in the core layer and it becomes smaller as it approaches layers near the mold wall”.

【R】Thanks for pointing out error. It has been corrected to “As he maximum size of the cell usually appears in the core layer and it becomes smaller as it approaches layers near the mold wall”.